# Effect of the Texture of the Ultrafine-Grained Ti-6Al-4V Titanium Alloy on Impact Toughness

**DOI:** 10.3390/ma16031318

**Published:** 2023-02-03

**Authors:** Iuliia M. Modina, Grigory S. Dyakonov, Andrey G. Stotskiy, Tatyana V. Yakovleva, Irina P. Semenova

**Affiliations:** Laboratory of Advanced Technologies, “Higher Engineering School of Aerospace Technologies” Center, Ufa University of Science and Technology, 32 Zaki Validi St., 450076 Ufa, Russia

**Keywords:** titanium alloy, ultrafine-grained structure, severe plastic deformation, texture, mechanical properties, impact toughness

## Abstract

In this work, the strength properties and impact toughness of the ultrafine-grained (UFG) Ti-6Al-4V titanium alloy produced by severe plastic deformation (SPD) in combination with upsetting were studied, depending on the direction of crack propagation. In the billets processed by equal-channel angular pressing (ECAP), the presence of anisotropy of ultimate tensile strength (UTS) and ductility was observed, conditioned by the formation of a metallographic and crystallographic texture. At the same time, the ECAP-processed UFG alloy exhibited satisfactory values of impact toughness, ~0.42 MJ/m^2^. An additional upsetting of the ECAP-processed billet simulated the processes of shape forming/die forging and was accompanied by the development of recovery and recrystallization. This provided the “blurring” of texture and a reduction in the anisotropy of UTS and ductility, but a difference in impact toughness in several directions of fracture was still observed. It is shown that texture evolution during upsetting provided a significant increase in the crack propagation energy. The relationship between microstructure, texture and mechanical properties in different sections of the material under study is discussed.

## 1. Introduction

Two-phase titanium alloys, in particular, Ti-6Al-4V, are widely applied as structural materials in aircraft engine building owing to their high specific strength and corrosion resistance. Operation under high static and dynamic loads, as well as the rapid development of this industry, imply ever stricter requirements towards the structural strength of materials, as well as an increase in their reliability and performance properties. Traditionally, strengthening of two-phase titanium alloys is achieved by thermomechanical treatment by means of controlling phase structural transformations, which have largely exhausted their potential. One of the promising approaches for increasing the strength of metals and alloys, as well as their fatigue endurance, is the formation of an ultrafine-grained (UFG) structure by severe plastic deformation (SPD) [1,2,3,4,5,6,7]. Besides, this approach to two-phase titanium alloys provides a high-strain-rate and/or low-temperature super-plasticity [8,9], which makes it possible to implement the process of shape forming of parts by isothermal forging at lower temperatures of 750–780 °C as compared to 900–940 °C for the conventional technology [10].

Crack propagation resistance or fracture toughness, which is one of the main characteristics providing the structural strength and reliability of a material, is known to be a structurally sensitive parameter. In many cases, high strength and fracture toughness are mutually exclusive characteristics in metals and alloys [11,12,13]. Therefore, an increase in fracture toughness can be achieved by creating special UFG structures corresponding to a higher crack resistance. For example, in two-phase titanium alloys, the “hard” phase could be the α-phase with a hexagonal close-packed (hcp) lattice where the main strengthening mechanisms are realized, and the “soft” phase could be the β-phase with a body-centered cubic (bcc) lattice that conditions a large number of slip systems [14]. Structural parameters should also be taken into consideration in UFG alloys, such as the phase shape and size, the precipitation of secondary phases and the state of grain boundaries, which influence the material’s capacity for strain hardening and normally determine the ductile properties and fracture toughness of UFG materials [11,15,16]. The UFG structure formation may also be accompanied by the formation of multi-component metallographic and crystallographic textures. Heavily textured materials often exhibit the anisotropy of mechanical properties [17,18,19,20,21]. For example, according to [21], a highly textured material from the Ti6/4 alloy exhibited essentially different strength characteristics depending on the test direction with respect to the predominant texture, where loading in the transverse direction, perpendicular to the basal planes, contributed to a high ultimate tensile strength (UTS) and yield strength (YS), while loading under cyclic loads in the longitudinal orientation provided a high fatigue endurance limit. In [11,22,23], the authors examined the effect of grain morphology and orientation with respect to the fracture direction during impact toughness and fracture toughness tests; it was found that, in the case of fracture along elongated grains, much less energy was required for the formation of both microcracks and pores in front of the main crack tip. In addition, in [11], through the example of Armco-iron subjected to equal-channel angular pressing (ECAP), an assumption was made that crystallographic texture appears less significant for the emergence of anisotropy in the fracture behavior than grain texture (grain alignment). However, in two-phase Ti alloys, predominantly with an hcp lattice, the number of slip systems for providing deformation is much smaller, and therefore the contribution of crystallographic texture may have a great significance for crack propagation along or across the oriented texture.

Therefore, the aim of this work is to study the effect of the metallographic and crystallographic texture produced in a UFG billet and its heredity during the subsequent deformation by upsetting on the mechanical properties and impact toughness of the UFG Ti-6Al-4V titanium alloy.

## 2. Materials and Methods

In this work, we used the (α + β) two-phase titanium alloy Ti-6Al-4V in the form of a rod with a diameter of 20 mm (manufactured by VSMPO-AVISMA, Verkhnyaya Salda, Russia). The initial billets were subjected to a preliminary heat treatment (HT) via the following regime: water quenching at a temperature of 960 °C followed by annealing for 4 h at a temperature of 675 °C with air cooling. The microstructure of the two-phase (α + β) Ti-6Al-4V titanium alloy in the coarse-grained (CG) state (Figure 1a) after heat treatment represented a globular primary α-phase ($\alpha_{P}$) with a size of 5 μm and a lamellar mixture of the α- and β-phases. The volume fraction of the primary α-phase was 50%. 

A UFG state was produced by severe plastic deformation: a billet with a length of 100 mm (in the direction along the X axis, Figure 2a) was subjected to 4 passes of equal-channel angular pressing (ECAP) with a channels intersection angle of 120° (Figure 1b), fixed in the die, via route C at a temperature of 700 °C, i.e., with 180° rotation after each pass along the longitudinal axis (0°–180°–0°–180°). The true strain introduced into the material was 2.8. To simulate the process of shape forming of a part, the upsetting of the billets with a UFG structure was conducted in the direction along the Z axis at a temperature of 750 °C (ε = 30%). Figure 2 shows the principle of cutting out samples from the produced billets with a UFG structure for further study.

Electron microscopic studies of the samples were conducted on a JEOL JSM 6390 (JEOL, Tokyo, Japan) scanning electron microscope (SEM) at an accelerating voltage of 20 keV and a JEOL JEM 2100 (JEOL, Tokyo, Japan) transmission electron microscope (TEM) at an accelerating voltage of 200 kV. The X-ray phase analysis (XPA) was carried out using a D8 Discover Bruker X-ray diffractometer with monochromatic copper radiation (Bruker AXS GmbH, Karlsruhe, Germany). The diffraction spectra were processed using a licensed software for the D8 Discover Bruker diffractometer (Bruker AXS GmbH, Karlsruhe, Germany) that enabled calculating the mass fraction of phases from the intensity of the recorded X-ray diffractions [24]. Analysis of the crystallographic texture was conducted by recording incomplete direct pole figures (DPFs) via the tilt method [25,26] with the use of reflections for the α-phase: (10.0), (10.2), (11.0), (20.2), (11.4). The complete DPFs (0001), {11.0} and {10.0} were restored using the LaboTex software Version 3.0 (LaboSoft s.c., Krakow, Poland) [27]. The tensile mechanical tests of the flat samples cut out from the billets after ECAP and ECAP + upsetting along the directions of the axes X, Y and Z (Figure 2), with a gauge section of 0.5 × 1.0 mm and a gauge length of 4 mm, were conducted at room temperature with a rate of $10^{-3}$ $s^{-1}$ on an Instron 5982 universal testing machine (Instron Engineering Corporation, Buckinghamshire, UK). The force measurement accuracy was 1%. Three samples were tested for each state. The impact toughness tests of the standard samples with a size of 10 × 10 × 55 mm, cut out in the longitudinal section (in the direction of the X axis) of the billets after ECAP and ECAP + upsetting, with a V-shaped notch, were conducted at room temperature using an Instron CEAST 9350 (Instron Engineering Corporation, Buckinghamshire, UK) drop weight impact testing machine. Besides, the impact toughness tests of the samples after ECAP + upsetting were carried out in two directions in the YZ plane (Figure 2b): in the direction of upsetting (along the Z axis) and perpendicularly to the direction of upsetting (along the Y axis). At least three samples were used per experimental point.

## 3. Results 

During the equal-channel angular pressing of the Ti-6Al-4V alloy the primary globular α-phase was fragmented, and the development of the globularization of the lamellar structure provided the formation of fine grains/sub-grains of the α-and β-phases having an equiaxed shape with a mean size of 0.4 μm. It is visible that the primary α-phase is elongated at an angle of approximately 10 to 30° with respect to the Z axis (Figure 3a), which indicates its predominant metallographic orientation.

An additional deformation by upsetting resulted in the development of recrystallization and a certain increase in the size of the globularized α- and β-grains to 0.5–1 μm. Individual grains of the primary α-phase were fragmented during deformation and acquired clearer boundaries due to the development of recrystallization (Figure 3b).

The X-ray phase analysis revealed an increase in the fraction of the β-phase as a result of the deformation by ECAP and ECAP + upsetting (Table 1). Such a change in the phase composition is characteristic of this titanium alloy when processed at elevated temperature [28]. A large fraction of the α-phase, which has an hcp lattice, may exhibit considerable anisotropy of mechanical properties depending on the texture formed during processing [21,29,30].

According to the obtained data, the texture of the ECAP-processed α-phase looks typical for the lateral surface of a rod passing through a channel. The lateral surface is perpendicular to the Y axis (see Figure 4a). It can be seen in the figure that the normals are rotated in the XZ plane, which has been described multiple times in [31,32] and is already regarded as a classic explanation for the formation of a crystallographic texture in the shear plane [33], the inclination of which is determined by the channels mating angle. In this case, this angle is about 10° with respect to the Z axis.

The change in the texture resulting from the upsetting of the billets (Figure 4b) leads to the reorientation of the basal normal in the direction of the compression axis by angles of 85 and 57°, which provides a significant decline in pole density at an angular distance of 70–90° from the compression axis Z. Basal slip contributes to the smooth reorientation of the basal normals in the direction of the compression axis. Regularities in texture formation during plastic deformation through the example of Zr alloys with a hexagonal lattice are considered in detail in [34].

Figure 5 shows the typical curves from the tensile mechanical tests. It can be seen that the formation of a UFG state by ECAP increases strength on average by up to 15%, while subsequent upsetting increases strength by up to 10% in the longitudinal direction (XY) with respect to the initial coarse-grained state. In the samples of the ECAP-processed UFG Ti-6Al-4V alloy, there was a pronounced anisotropy of mechanical properties produced in samples from different billet sections. The maximum differences between the UTS and YS values in the sections of the ECAP-processed billet are 8 and 13%, respectively, while uniform elongation in the billet’s cross section decreases three-fold with respect to the longitudinal direction. The subsequent upsetting of the ECAP-processed UFG alloy decreases the anisotropy of strength and increases ductility.

The impact toughness test results for the Ti-6Al-4V titanium alloy in different structural states are shown in Table 1. The impact toughness value for the coarse-grained state was 0.49 $\mathrm{MJ}/m^{2}$. The UFG state formation by ECAP reduces impact toughness to 0.42 $\mathrm{MJ}/m^{2}$. The selection of notch cutting and fracture direction in the transverse section is not important, since deformation by ECAP enables producing a rather homogeneous microstructure [5]. The additional upsetting enables preserving impact toughness at a level of 0.41 $\mathrm{MJ}/m^{2}\;$in the test direction perpendicular to the billet compression axis, while impact toughness in the test direction along the compression axis declines to 0.32 $\mathrm{MJ}/m^{2}$. 

## 4. Discussion

In order to evaluate the effect of structure and texture on the mechanical properties and impact toughness, we analyzed the microstructure after ECAP and ECAP + upsetting in the sections YZ and XZ. A decrease in the size of the α- and β-phase grains in the ultrafine-grained state resulted in an increase in yield strength compared to the coarse-grained alloy, which agrees with the known Hall-Petch relationship [35]. At the same time, the tensile curves of the UFG samples exhibited early strain localization during tension and a decrease in the uniform elongation $\delta_{u}$ from 4.5 to 2.0%. This is conditioned by the fact that in UFG metals with a grain size below 1 μm the nucleation of new dislocations and their accumulation are impeded, which may lead to early strain localization and fracture [36]. The pronounced anisotropy of tensile mechanical properties (Table 1, Figure 5) of the UFG Ti alloy after ECAP may be associated with the metallographic and crystallographic texture that is inherent in a material after deformation. An additional deformation by upsetting, due to the development of the recrystallization processes, leads to the “blurring” of the metallographic texture and a decrease in the anisotropy of mechanical properties. A certain increase in the grain/sub-grain size in the process of upsetting, as well as an increase in the volume fraction of the softer β-phase with a bcc crystalline lattice led to a slight decline in strength and an increase in ductility as compared to ECAP [28]. The evening out of the volume fraction of the β-phase in the billet sections in the process of upsetting also provides a more isotropic character of the produced UFG structure. Thus, the upsetting of the ECAP-processed samples of the Ti-6Al-4V titanium alloy, simulating the die forging/shape forming processes in this work, reduces the anisotropy of mechanical properties and provides a level of ductility comparable to the initial state, which is favorable for its technological application. 

Table 1 presents the results of the tests for impact toughness (KCV). It is apparent that for the UFG states the tendency is preserved for the KCV values to decrease with increasing strength characteristics and decreasing capacity for hardening [1,11,22], which depends on the uniform elongation $\delta_{u}$ and can also be defined as the $\sigma_{\mathrm{UTS}}/\sigma_{\mathrm{YS}}$ relationship [37]. As examined in [11,23], when the grains of the elongated primary α-phase are positioned perpendicularly to the crack growth direction, they inhibit its growth and branching, which provides a satisfactory level of impact toughness in the Ti-6Al-4V titanium alloy in the UFG state produced by ECAP, having a higher strength compared to its coarse-grained state. When total impact energy (*A*) is divided into crack initiation work (*A_i_*) and crack propagation work (*A_p_*), it can be noted that the main contribution into facture resistance in the ECAP-processed UFG state during impact toughness testing is made by crack generation work, which is maximal among all the presented states and amounts to 27.9 J, which is conditioned by a high strength value; meanwhile, the crack propagation work is minimal and amounts to 4.5 J, since crack propagation is directly proportional to the deformation capacity (Table 1) [1,11,22]. It should be noted that the impact toughness and uniform elongation values of the samples after ECAP and ECAP + upsetting, tested in the Y direction, are comparable between each other. However, the UTS values in these samples were noticeably different and the highest UTS value of ~1248 MPa was exhibited by the ECAP-processed state. The increased values of strength and a good impact toughness of the ECAP-processed samples are evidently conditioned by textural features and grain elongation perpendicular to the crack propagation direction [11], as well as the state of boundaries, whereas during the subsequent upsetting the impact toughness in the fracture direction perpendicular to the compression axis (Figure 2b) is conditioned by an increase in the fraction of high-angle boundaries in the structure due to the recovery and recrystallization processes [38], while the texture “blurring” and an increase in grain size lead to a decline in strength to 1090 MPa. The fragmentation of the elongated primary α-phase into a more homogeneous globularized one in the process of upsetting should lead to a decrease in the impact toughness value since the crack path passing through equiaxed phases is shorter than that passing through lamellar phases [39]. During the impact toughness tests of the samples after ECAP + upsetting in the direction perpendicular to the upsetting direction, the decline in strength characteristics, in comparison to the UFG state after ECAP, conditions a certain decrease in the crack initiation work to 26.4 J. However, the blurring of crystallographic texture leads to a larger inclination of the basal planes with respect to crack propagation (Figure 4 and Figure 6), which increases the crack propagation path, since the crack propagates via the energetically favorable grain orientation axes, resulting in the need for a higher energy for crack propagation—the crack propagation energy increased from 4.5 J after ECAP to 7.0 J (Table 1). The impact toughness in the direction of fracture along the compression axis decreases to 0.32 $\mathrm{MJ}/m^{2}$ while the capacity for strain hardening in this direction simultaneously increases, which is an unusual result and may be attributed to textural features in the UFG material after upsetting (Figure 4 and Figure 6). A considerable decline in pole density at an angular distance of 70–90° from the compression axis (Figure 4) leads to a higher value of impact toughness in the direction of fracture opposite to the compression axis, 0.41 $\mathrm{MJ}/m^{2}$. Figure 6b shows that the orientation of the c-axes of the hcp lattice with the Z direction (crack propagation direction) makes up an angle close to 90°. Taking this into consideration, in can be presented that deformation along Z is provided by an easy prismatic slip, therefore the crack propagation energy in this case is relatively low/decreases to 6.0 J (Table 1). In the case when the impact toughness test direction was perpendicular to the upsetting direction, the crack propagation direction was almost parallel to the c-axis of the hcp lattice. As known, the deformation of the hcp lattice along the c-axis is impeded and may be realized due to pyramidal c + a slip (with higher critical resolved shear stress) or deformation twinning that is not active in Ti-6Al-4V. Taking this into consideration, it can be assumed that crack development in this case requires a higher energy. In its path, the crack is forced to bend and change direction to develop via the more favorable orientations, which corresponds to a higher value of crack propagation energy—7.0 J (Table 1). A decline in the strength characteristics also reduces the crack initiation energy to 20.4 J (Table 1). 

Thus, it is evident that, during the shape forming of a part, the transformation of the UFG structure and texture occurs due to the development of the dynamic recrystallization and recovery processes, and the “blurring” of the metallographic and crystallographic texture conditions a decrease in its negative effect on mechanical properties and impact toughness. By means of selecting the processing regimes, it is possible to produce special UFG structures that may exhibit a combination of high strength and impact toughness. 

## 5. Conclusions

1.The formation of UFG states by ECAP results in a decrease in the grain size of the secondary α- and β-phases to 0.4 μm, which provides an increase in UTS to 1150 MPa in the billet’s longitudinal direction. The metallographic and crystallographic texture produced in the process of ECAP leads to a certain anisotropy of strength and ductility.2.The simulation of the shape forming process of the UFG material by upsetting leads to the development of recovery and recrystallization processes, an increase in the size of secondary phases to 0.5–1 μm and a decrease in UTS to 1095 MPa in the longitudinal direction. Texture “blurring” leads to a decline in the anisotropy of mechanical properties in the billet.3.The impact toughness of the Ti-6Al-4V titanium alloy with a UFG structure after ECAP is 0.42 $\mathrm{MJ}/m^{2}$. An additional upsetting leads to a decrease in impact toughness to 0.32 $\mathrm{MJ}/m^{2}$ in the compression direction, but a decline in pole density at an angular distance of 70–90° from the compression axis conditions the preservation of impact toughness at a level of 0.41 $\mathrm{MJ}/m^{2}$.4.The effect of crystallographic texture on impact toughness is shown. Texture blurring during upsetting provided an increase in the crack propagation energy. The crack propagation in the Y direction—almost parallel to the axis with an HCP lattice—was impeded due to the absence of easy slip systems. Therefore, the crack is forced to bend and change its direction via more favorable orientations, which resulted in the highest crack propagation energy value of 7.0 J.

## Figures and Tables

**Figure 1 materials-16-01318-f001:**
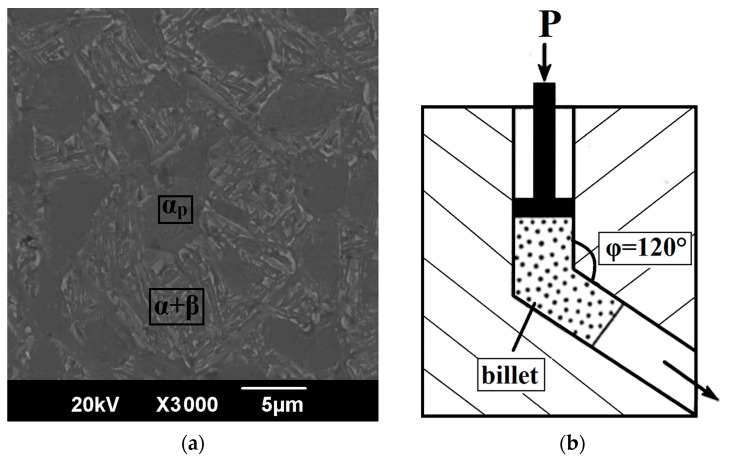
Equal-channel angular pressing: (**a**) microstructure of the billet; (**b**) principle of ECAP.

**Figure 2 materials-16-01318-f002:**
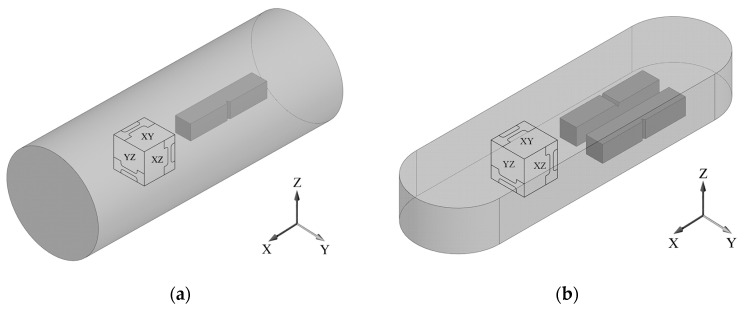
Schematic view of the Ti-6Al-4V titanium alloy samples after: (**a**) ECAP; (**b**) ECAP + upsetting, for microstructural studies, tensile mechanical tests and impact toughness tests with a V-notch.

**Figure 3 materials-16-01318-f003:**
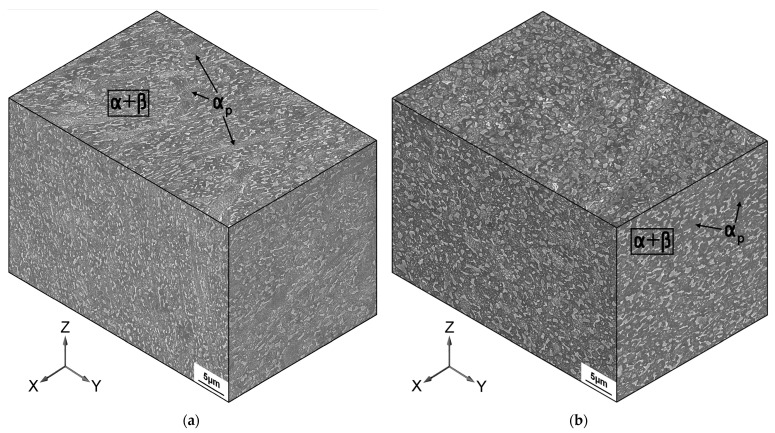
Microstructure of the Ti-6Al-4V titanium alloy: (**a**) ultrafine-grained (UFG) state after ECAP; (**b**) UFG state after ECAP + upsetting.

**Figure 4 materials-16-01318-f004:**
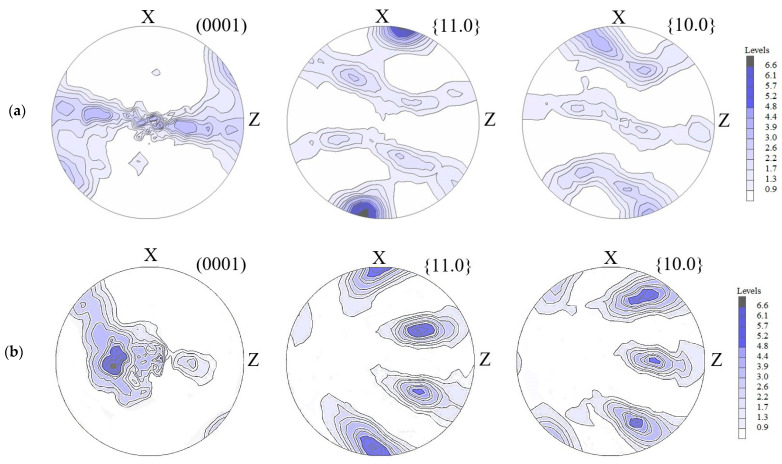
(0001), {11.0} and {10.0} DPFs of the α-phase in the XZ-section for (**a**) the ECAP-processed billet and (**b**) the ECAP-processed billet after upsetting along the Z direction. Outer directions are indicated near the DPFs.

**Figure 5 materials-16-01318-f005:**
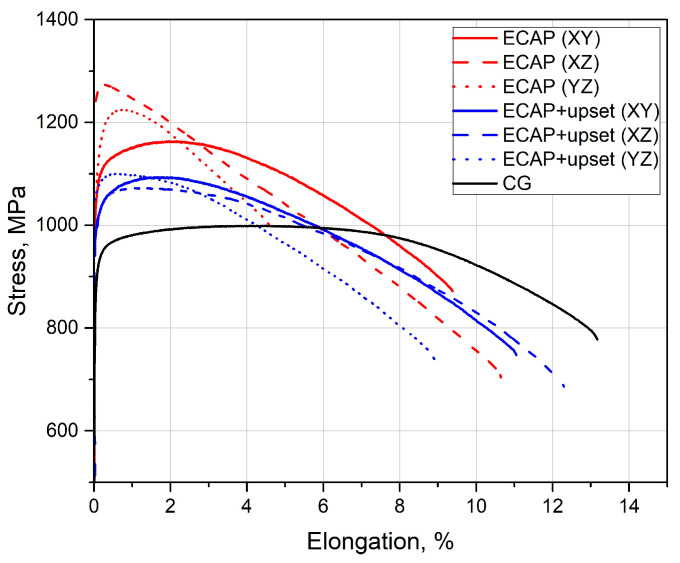
Typical curves from the tensile mechanical tests of the Ti-6Al-4V titanium alloy.

**Figure 6 materials-16-01318-f006:**
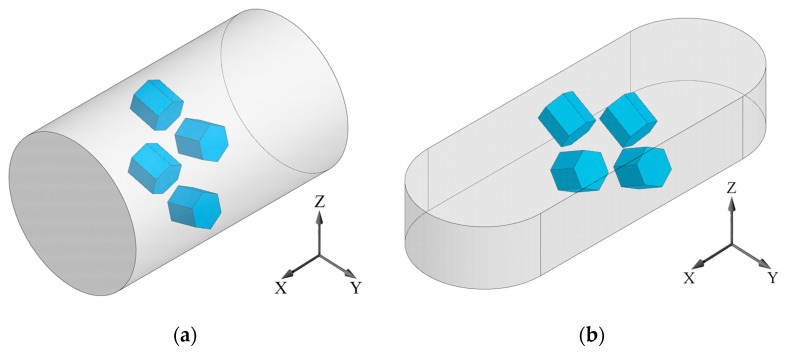
Schematic view showing the position of the crystallites of the hcp α-phase in the UFG structure of the Ti-6Al-4V titanium alloy after: (**a**) ECAP; (**b**) ECAP + upsetting.

**Table 1 materials-16-01318-t001:** Mechanical properties of the Ti-6Al-4V alloy.

State	Vα-Ti Phase, % wt.	V β-Ti Phase, % wt.	$\sigma_{\mathrm{UTS}},\;\mathrm{MPa}$	$\sigma_{\mathrm{YS}},\;\mathrm{MPa}$	δ,%	$\delta_{u},\;\%$	KCV, $\mathrm{MJ}/m^{2}$	Impact Energy, *A*, J	*A_i_*, J	*A_p_*, J
Coarse-grained (CG)	90	10	990 ± 20	920 ± 20	13.0 ± 1.0	4.5 ± 0.5	0.49 ± 0.03	39.0 ± 1.0	15.0 ± 1.5	24.0 ± 1.0
UFG ECAP (XY)	79	21	1150 ± 17	1100 ± 15	10.1 ± 1.9	2.0 ± 0.5	-			
UFG ECAP (XZ)	80	20	1252 ± 22	1250 ± 24	10.4 ± 1.1	0.3 ± 0.1	-			
UFG ECAP (YZ)	85	15	1248 ± 26	1220 ± 36	4.5 ± 1.3	0.6 ± 0.1	0.42 ± 0.05	32.4 ± 0.5	27.9 ± 1.5	4.5 ± 1.5
UFG ECAP + upsetting (XY)	80	20	1095 ± 17	1030 ± 28	11.5 ± 1.4	1.9 ± 0.2	-			
UFG ECAP + upsetting (XZ)	78	22	1080 ± 9	1040 ± 15	12.2 ± 1.5	1.1 ± 0.3	0.32 ± 0.04(along the Z axis)	26.4 ± 2.5	20.4 ± 1.0	6.0 ± 2.7
UFG ECAP + upsetting (YZ)	82	18	1090 ± 21	1065 ± 27	9.0 ± 0.1	0.7 ± 0.2	0.41 ± 0.10(along the Y axis)	33.4 ± 5.5	26.4 ± 5.2	7.0 ± 2.3

## Data Availability

Not applicable.

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
