# Peer review of "Effect of the Texture of the Ultrafine-Grained Ti-6Al-4V Titanium Alloy on Impact Toughness"

_materials, 2023, doi:10.3390/ma16031318_

Round 1
Reviewer 1 Report
The tensile properties and impact toughness of UFG Ti-6Al-4V alloy was presented and discussed, however, the present form is not enough for publication.
1. The present title can not reflect the main content;
2. Review of the reported relation between mechanical property and texture of Ti and Ti alloys is lack and should be given to show the novelty and necessity of the study;
3. The detailed process for ECAP should be given;
3. Fig.5 is not clear enough for the mechanism. The discussion of texure influnce on impact toughness is too simple, necessary correlation should be profoundly discussed, especially for the obvious decreasing of impact toughness to 0.32;
4. In table 1, for UFG ECAP(YZ), the phase composition is similar with the initial biliet, why?
5. In conclusions, 0.31 should be 0.32;
Reviewer 2 Report
Comments and Suggestions for Authors
Abstract
1. A modest suggestion, the presentation in this section must be perfect and substantial. What gap do you want to fill? What is your contribution? What is the value of your work to other researchers?
2. Typically, we do not abbreviate professional phrase when they first appear. See line 8, 9, 10…
Results
1. Line 98 and Table 1. Which result of phase fraction is correct? 50 or 90?
Discussion
1. Line 98 and Table 1. Which result of phase fraction is correct? 50 or 90?
2. Line 215. According to Table 1, it seems like the impact toughness of ECAP-processed UFG (0.41) does not have a notable enhancement comparing to traditional UFG sample, which means the “blurring” of the metallographic and crystallographic texture does not decrease negative effect on impact toughness. In addition, traditional UFG exhibits anisotropic elongation which depends on XYZ directions. Perhaps the “blurring” strategy could reduce this anisotropic elongation.
I recommend a minor revision.

Reviewer 3 Report
The current manuscript is a study on the effect of crystallographic texture on the mechanical performance of ultra-fine grained Ti-6Al-4V alloy. The tests are properly designed.
A pity is that the physical reasons of the texture evolution were not deeply discussed.
Furthermore, the explanation of the difference between the mechanical performances of the ECAP and ECAP+upsetting materials is based on the morphology (Fig. 5) but not the texture of the grains.
It is suggested to have a unified coloring scale of Fig.3 (a) and (b) so that the readers can compare the intensity between the two cases in a clearer way.
Reviewer 4 Report
In my opinion, the manuscript is an extension of two works (references 34 and 36) published by the authors.
The difference consists in including some samples for tensile and impact tests at different directions from that of published works in order to consider crystallographic texture as stated in the title of the manuscript.
The different behaviors shown in the tests results are poorly discussed in my opinion, basically explained by using references, e. g. refs 34 and 36.
Explanations are mostly attributed in a general way to upsetting, textural features and blurring effect, such as in:
- lines 191-193: "The increased values of strength and a good impact toughness of the ECAP-processed samples are evidently conditioned by textural features..."
- lines 173-175: "An additional deformation by upsetting, due to the development of the recrystallization processes, leads to the “blurring” of the metallographic texture and a decrease in the anisotropy of mechanical properties."
- lines 198-199: "...texture “blurring” and an increase in grain size lead to a decline in strength to 1090 MPa."
In other words, there is not a specific discussion for each mechanical behavior for the samples at each direction. This is the essential part and the contribution of the manuscript that the authors must provide much improvement.
Round 2
Reviewer 1 Report
All the key points are well modified, however, some minor modification still needed.
1. In Table 1, the phase content is not right for alpha-Ti phase for UFG ECAP(YZ);
2. It is suggested to include the main discussion results for the texture-impact relation in the abstract and conclusion part;
Author Response
Dear reviewer, thank you for your time. Our responses are below.
All the key points are well modified, however, some minor modification still needed.
- In Table 1, the phase content is not right for alpha-Ti phase for UFG ECAP(YZ);
We have re-checked the data and made changes in Table 1.
- It is suggested to include the main discussion results for the texture-impact relation in the abstract and conclusion part;
The corresponding corrections have been made (Line 18-19, 283 - 288).
Reviewer 4 Report
The authors have satisfactorily addressed all my previous comments.
Author Response
Dear reviewer, thank you for your time.